# Impact of Diet and Exercise Behaviors on Body Mass Index of Advanced Practice Nurses in the United States

**DOI:** 10.3390/nu17233654

**Published:** 2025-11-22

**Authors:** Melissa J. Benton, Sherry J. McCormick, Natasha Smith-Holmquist, Deborah Tuffield

**Affiliations:** Helen & Arthur E. Johnson Beth-El College of Nursing & Health Sciences, University of Colorado Colorado Springs, Colorado Springs, CO 80918, USA; smccorm3@uccs.edu (S.J.M.); nsmithho@uccs.edu (N.S.-H.); dtuffiel@uccs.edu (D.T.)

**Keywords:** overweight, obesity, body mass index, exercise, fruit, vegetables, protein

## Abstract

**Background/Objectives**: Advanced Practice Nurses (APNs) counsel patients regarding diet and exercise behaviors and serve as role models for health promotion and prevention of chronic disease. This study evaluated personal diet and exercise behaviors of APNs and their association with body mass index (BMI) as a biomarker of obesity and disease risk. **Methods**: APNs (N = 1268) self-reported height and weight, and answered four questions regarding personal diet and exercise. Based on BMI, they were grouped as normal weight (≤24.9 kg/m^2^) and overweight/obese (≥25.0 kg/m^2^). **Results**: The prevalence of overweight/obesity was 55%. The majority of APNs engaged in muscle strengthening exercises (53%) and consumed fruits and vegetables (62%), and protein foods and/or supplements (94%), while less than half engaged in moderate–vigorous physical activity (46%). Exercise behaviors (moderate–vigorous physical activity and muscle strengthening exercises) had a statistically significant impact on BMI. The predicted decrease in BMI due to participation in moderate–vigorous physical activity was 2.06 kg/m^2^ and the predicted decrease due to muscle strengthening exercises was 1.35 kg/m^2^. Diet behaviors (consumption of fruit, vegetables, and protein) were not found to have a significant impact on BMI. **Conclusions**: The prevalence of overweight/obesity among APNs in the United States is less than what is reported for the general adult population. Exercise behaviors, especially moderate–vigorous physical activity, significantly impact BMI and are associated with clinically meaningful differences. By comparison, healthy diet behaviors, including consumption of fruits, vegetables, and protein, are relatively widespread among advanced practice nurses but do not appear to significantly impact BMI.

## 1. Introduction

The prevalence of obesity is increasing globally [1]. In 2005, 396 million adults were obese, representing 10% of the world’s population, and the number was estimated to increase to more than one billion by 2030 [2]. Obesity rates vary between countries [3], but recent data from the United States indicate that obesity among adults exceeds 40% [4,5]. Nurses are also at risk, with current data reflecting an overall 16% prevalence of obesity among nurses in 29 countries [6]. Obesity, measured as a body mass index (BMI) greater than 30 kg/m^2^, is positively associated with mortality and can shorten life by as much as 9 years [7]. Furthermore, more than 4% of all adult deaths are attributable to obesity [7]. It is well established that obesity is related to poorer health outcomes [7,8]. These include type 2 diabetes, cardiovascular disease, cancer, sleep apnea, kidney disease, and musculoskeletal disorders [9,10,11]. Furthermore, although evidence is limited, an association has been identified between obesity and risk of self-harm [7]. Mortality increases approximately 20% with every 5 kg/m^2^ increase in BMI [7], and is highest for those with a BMI greater than 40 kg/m^2^ [12]. By comparison, the lowest mortality risk is consistently observed for adults with a BMI between 18.5–24.9 kg/m^2^ [12].

Maintaining a healthy BMI and participating in healthy lifestyle behaviors can decrease mortality risk and increase life expectancy [13], but national and international trends demonstrate decreased adherence over time to healthy diet and physical activity behaviors among adults [14,15,16]. Diet and exercise interventions that effectively decrease BMI either alone or combined with caloric restriction include consumption of fruits and vegetables, consumption of protein, moderate to vigorous aerobic training, and resistance training [17,18,19,20]. Unfortunately, adherence to these healthy behaviors is poor, and between 14% and 18% of adults report no participation in any diet or exercise behaviors [21,22]. Nurses also report low participation in weight management behaviors and the inability to manage personal obesity, despite knowledge of the negative health outcomes [23,24]. Generally, less than 50% of nurses are physically active and consume a healthy diet [25], while less than 20% report daily exercise, less than 40% report exercise 2–5 days per week, and less than 50% report consumption of 5 servings of fruit and vegetables daily [26,27].

Advanced nursing practice is founded on advanced education in nursing, with the minimum expectation of a master’s degree for entry into practice [28]. Globally, advanced practice nursing is expanding, with more than two dozen countries recognizing advanced practice roles [29]. Although there are variations in regulation, credentialing, and titling between countries [29], clinical leadership and patient education are fundamental to the advanced practice role [28]. As clinicians and leaders, Advanced Practice Nurses (APNs) counsel patients regarding diet and exercise behaviors and serve as role models for health promotion and prevention of chronic disease. Role modeling is not only a fundamental part of clinical education [30], but it can also increase self-efficacy for behaviors such as physical activity [31]. However, role models teach implicitly by example [30,32], so their personal characteristics and behaviors can be influential not only on their personal health outcomes [33], but also on the behaviors of others [34].

The Health Belief Model provides a theoretical foundation for health behavior change based on the premise that individuals are likely to engage in health-promoting behaviors if they perceive that they are susceptible to a health problem, that the problem has potentially serious health consequences, that initiating specific behaviors can reduce risk, and they have confidence (self-efficacy) that they can successfully perform those behaviors [35]. APNs have knowledge and understanding regarding the health risks of obesity and the potential benefits of diet and exercise behaviors [23,36], which may predispose them to regularly engage in these behaviors. However, evidence is not clear whether APNs engage in the healthy behaviors needed to maintain a healthy body weight and promote their own health outcomes. A better understanding of their personal diet and exercise behaviors is needed. Therefore, this study aimed to evaluate personal diet and exercise behaviors of APNs and their association with BMI as a biomarker of obesity.

## 2. Materials and Methods

### 2.1. Design, Setting, and Sample

This was a cross-sectional study designed to collect data one time with minimal participant burden. The study used a convenience sample of APNs who completed an online survey (Qualtrics, Provo, UT, USA) regarding their personal diet and exercise behaviors and self-reported their height and weight for calculation of BMI. Study recruitment was conducted through postings on social media sites on Facebook, flyers sent to professional organizations with requests for dissemination, and emails to professional colleagues providing information about the study. A snowball sampling strategy was used. Participants were included if they affirmed that they were at least 18 years of age and practiced at least one day a week in one of the four advanced practice roles (nurse anesthetist, nurse-midwife, clinical nurse specialist, nurse practitioner) licensed to practice in the United States [28,37]. The single exclusion criterion was failure to answer one or more of the survey items.

The Institutional Review Board of the University of Colorado Colorado Springs approved this study as exempt (approval #2022-147) and all participants affirmed consent and voluntary participation prior to accessing the online survey. They were informed that all questions were optional, and they could choose to close and exit the survey at any time.

### 2.2. Measurement

Participants answered three questions regarding their age, gender, and race/ethnicity. The responses to these questions were used solely to describe the population. Because this study was conducted in the United States, participants were asked to self-report their height in feet and inches and body weight in pounds (imperial system). Researchers then converted measurements to metric values (height in meters and weight in kilograms) for calculation of BMI (kg/m^2^).

The survey questions regarding diet and physical activity were developed by the researchers based on the Dietary Guidelines for Americans [38] and Physical Activity Guidelines for Americans [39], and were not validated or pilot tested prior to the study. Participants were asked two questions regarding dietary intake and two questions regarding exercise. Dietary intake questions were worded as: (1) *On average, do you eat at least 5 servings of fruit and/or vegetables a day?* and (2) *On average do you eat protein foods and/or supplements daily?* Exercise questions were worded as: (1) *On average, do you engage in moderate–vigorous physical activity for at least 30 min on 5 or more days of the week? Moderate–vigorous activity is at an intensity that slightly increases your heart rate or breathing and makes it somewhat difficult to carry on a conversation*, and (2) *On average, do you engage in muscle strengthening activities on at least 2 or more days of the week?* Responses were dichotomized as either yes or no.

### 2.3. Statistical Analysis

Data were analyzed using SPSS version 29 (IBM Corp. Armonk, NY, USA). To decrease the chance of a type I error to less than 1%, statistical significance was pre-determined to be *p* ˂ 0.01. Participants were described using descriptive statistics, including means ± standard deviations for continuous variables and frequencies (percents) for categorical variables. The dependent variable (BMI) was found to be non-normally distributed so a Mann–Whitney U test was used to compare differences between participants grouped as normal weight (BMI ≤ 24.9 kg/m^2^) and overweight/obese (BMI ≥ 25.0 kg/m^2^). Significant relationships between participant BMI and diet and exercise behaviors were then identified using Spearman correlations (*r_s_*) and subsequently any correlations meeting the pre-determined level of significance were entered into a multiple regression model to determine their contribution to BMI. Assumptions for linearity, homoscedasticity, and normality were met [40].

## 3. Results

### 3.1. Characteristics of the Sample

Full data were available for 1268 advanced practice nurses (Table 1). Gender was self-reported as predominantly female (97%). Race/ethnicity was predominantly white (90%), followed by Hispanic (5%), Black (3%), and Asian/Pacific Islander (3%). Average age was 46.7 ± 11.2 years, and average BMI was 26.6 ± 5.4 kg/m^2^. The overall prevalence of overweight/obesity (BMI ≥ 25.0 kg/m^2^) was 55%. Although less than half (46%) reported engaging in moderate–vigorous physical activity, the majority of participants reported engaging in muscle strengthening exercises (53%), consuming fruits and vegetables (62%), and consuming protein foods and/or supplements (94%).

### 3.2. Comparison of Normal Weight and Overweight/Obese Participants

When groups were compared (Table 2), overweight/obese participants were significantly older than normal weight participants (47.5 ± 11.2 vs. 45.7 ± 11.0 years, *Z* = −2.88, *p* = 0.004). Also, statistically significant between-group differences were observed for exercise behaviors. Significantly more normal weight participants reported engaging in moderate–vigorous physical activity (*Z* = −6.59, *p* ˂ 0.001) and muscle strengthening exercises (*Z* = −5.81, *p* ˂ 0.001) than overweight/obese participants. However, although differences in fruit and vegetable consumption were observed, they did not meet the pre-determined level of significance, and there was no difference between groups for consumption of protein foods and supplements.

### 3.3. Correlation Between BMI and Diet and Exercise Behaviors

Correlation analysis identified negative relationships between BMI and exercise participation and consumption of fruits and vegetables that met the pre-determined level of significance. Specifically, BMI was significantly related to moderate–vigorous physical activity (*r_s_* = −0.233, *p* ˂ 0.001), muscle strengthening exercise (*r_s_* = −0.206, *p* ˂ 0.001), and fruit and vegetable intake (*r_s_* = −0.084, *p* = 0.003). However, there was no significant relationship between protein intake and BMI.

### 3.4. Contribution of Exercise and Diet Behaviors to BMI

Based on the results of the correlation analysis, participation in moderate–vigorous physical activity and muscle strengthening exercises, and consumption of fruits and vegetables were entered as independent variables into multiple regression with BMI as the dependent variable (Table 3). Models were adjusted for age due to the significant difference between groups (Table 2). Regular participation in moderate–vigorous physical activity explained 6.3% of the variability in BMI (*F*(2,1242) = 42.01, *p* ˂ 0.001) and the addition of muscle strengthening exercises increased the contribution of exercise to 7.6% (*F*(3,1241) = 34.26, *p* ˂ 0.001). The model was not improved by the addition of dietary factors. When consumption of fruits and vegetables was added, the contribution to BMI was not significantly increased (*p* = 0.169). Based on the final regression model, the predicted decrease in BMI due to participation in moderate–vigorous physical activity was 2.06 kg/m^2^ and the predicted decrease due to muscle strengthening exercises was 1.35 kg/m^2^.

## 4. Discussion

To our knowledge, this is the first study to examine the impact of diet and exercise on the BMI of APNs who serve as patient educators and role models in the community. Our sample is generally representative of APNs in the United States who are 90% female, 80% White, and have an average age of 50 years [41]. Surprisingly, dietary factors did not have a significant influence on BMI despite evidence from meta-analyses that weight loss interventions that include fruit and vegetable consumption can reduce body weight by an average of 2.8 kg [18], and weight loss interventions that include high protein consumption can reduce BMI by an average of 1.86 kg/m^2^ [20]. Instead, the current study demonstrated that exercise has a greater impact on BMI. This finding appears to be consistent with previous evidence that physical activity accounts for 15–30% of daily energy expenditure, while the metabolic cost of ingestion of food accounts for approximately 10% [42]. Both moderate–vigorous physical activity and muscle strengthening exercise had a significant influence on BMI and explained more than 7% of the variance in BMI among participants. Although this may appear to be negligible, differences in BMI of 2 units have been determined to be clinically meaningful [43]. Furthermore, based on the average height and weight of our participants (Table 1), the predicted decrease in BMI of 2.06 kg/m^2^ due to participation in moderate–vigorous physical activity represents a difference in body weight of approximately 13 lb (5.9 kg). By comparison, the predicted decrease of 1.35 kg/m^2^ due to participation in muscle strengthening exercises represents a difference in body weight of approximately 8.5 lb (3.9 kg). Both weight differences exceed the 5% difference in body weight that has been deemed clinically meaningful [44]. Finally, given an estimated 20% increase in overall mortality associated with an increase in BMI of 5 kg/m^2^ [7], the decreases of 2.06 kg/m^2^ and 1.35 kg/m^2^ predicted by our regression model are potentially consequential for health outcomes.

The relative lack of effect of diet was surprising and may be related to the relatively widespread consumption of fruits and vegetables and almost universal consumption of protein foods and/or supplements reported by our participants. Not only did a much larger majority of participants report consumption of fruits and vegetables compared to those reporting exercise behaviors, but when participants were compared as normal and overweight/obese groups there were no between-group differences in consumption. A majority of both normal (65%) and overweight/obese (59%) participants reported consumption of fruits and vegetables. This phenomenon was even more pronounced in relation to protein consumption, which was ubiquitous, with over 90% of all participants reporting daily consumption of protein foods and/or supplements. Furthermore, when broken down into groups by BMI, there were no between-group differences and more than 90% of both normal and overweight/obese participants reported protein consumption. For comparison, approximately 70% of adults in the United States report daily consumption of fruits and 90% report daily consumption of vegetables [45], but only 10–12% meet the daily recommendations for both fruits and vegetables [46]. In contrast, protein intake among adults in the United States is universal with average daily intakes between 75–80 g [47]. Recognizing that the APNs who participated in our study are well educated, they are likely knowledgeable and adherent to nutrition recommendations regarding consumption of fruits and vegetables as well as protein foods [38].

Approximately 50% of nurses in the United States meet the criteria for overweight and obesity [6,27,48,49,50] and less than 50% are physically active and consume a healthy diet [27,50]. Over a decade ago, the need for role models to encourage healthy behaviors among registered nurses was identified [51], yet nurses in the United States remain at risk for poor health outcomes due to poor diet and inadequate physical activity [25]. Nurses themselves have identified lack of role models as a barrier to engaging in personal health promoting activities [52]. Advanced education is a predictor of role modeling [50], and nurse leaders that engage in healthy behaviors are effective role models that facilitate health promotion within organizations [53]. Advanced practice nurses not only have advanced degrees but are also educated as nurse leaders [28]. Hence, they are appropriate to serve as role models to promote healthy behaviors.

Consistent with the Health Belief Model, knowledge and self-efficacy are predictors for diet and exercise behaviors among adults in general [54] as well as health care professionals [55]. APNs may have greater knowledge and self-efficacy for dietary consumption of fruits, vegetables, and protein than for exercise behaviors and this influenced their behaviors. This is consistent with adherence to diet and exercise behaviors among community-dwelling adults in general, where adherence to fruit and vegetable consumption consistently exceeds adherence to physical activity guidelines [56]. Education can increase knowledge and self-efficacy for exercise and diet, and increased self-efficacy can lead to improved exercise and diet behaviors [54,57,58,59,60]. Currently, there is evidence that education regarding exercise and diet is absent from medical curricula [61,62]. We can find no similar studies for graduate nursing programs, although it seems likely that similar gaps exist.

Although the 55% prevalence of overweight/obesity observed among our participants exceeded the U.S. population prevalence for obesity calculated from NHANES data [4,5], it should be noted that we combined overweight and obese participants for analysis to account for the BMI cutoff point of 24.9 kg/m^2^ that reflects the lowest mortality risk [12]. When NHANES data for overweight and obesity are combined, U.S. population prevalence is actually 73% [63], which exceeds the prevalence among the APNs in our study. Consistent with this finding, participation in diet and exercise behaviors by APNs also exceeded rates reported for U.S. adults in general [21].

### 4.1. Strengths and Limitations

The major strength of our study is its large sample size, which supports the generalizability of our findings. The major limitation is our use of self-report for measurement of height and weight, as well as diet and exercise behaviors. Although adults tend to overreport height and underreport weight, recent data from women and men in the United States demonstrate good agreement between objectively measured and self-reported height and weight supporting the validity of self-report for these data [64]. Moreover, accuracy of self-reported height and weight is improved by educational level [64]. Based on the advanced education required for the participants in our study (minimum of a master’s degree as basic educational preparation for advanced practice), we believe that overall our sample reported accurate measures of height and weight that resulted in accurate calculation of BMI, which has a 90% specificity for identification of excess body fat [65]. Although BMI is not a direct measure of body fat, it provides a surrogate measure that is feasible and cost-effective for clinical use [66]. There is a strong correlation between body fat and BMI among adults in the United States [67]. Agreement varies by race and ethnicity [68], although in White/Caucasian populations, agreement is relatively good [69]. Specifically, at the same body fat percent, BMI values for non-White adults can vary by as much as 4.5 kg/m^2^ either above or below BMI values for White adults [70]. Furthermore, BMI cannot distinguish between fat and lean mass [71], which limits its use for assessment of body composition per se. This inability may be applicable to our participants who reported relatively widespread engagement in muscle strengthening exercise. It is possible that this had an unappreciated influence on BMI values, even with accurate self-report.

Although the items related to physical activity and dietary consumption were not previously validated, there is recognized measurement error (both under and overreporting) even with validated instruments such as the International Physical Activity Questionnaire and the Global Physical Activity Questionnaire [72,73]. Specifically, respondents have difficulty quantifying activity as durations and frequencies [74,75]. Similar limitations exist for dietary intake tools [76,77,78]. There is also an issue of timeliness inherent in any data collection. For example, the IPAQ can require as much as 15 min for completion [79], which is a burden and likely a barrier to survey research. For physical activity and dietary data, given the limitations (over and underreporting) surrounding the need for quantification, we chose to use terminology taken directly from the U.S. Physical Activity Guidelines [39] and the U.S. Dietary Guidelines [38] that are used for patient counselling by healthcare providers in the United States and which we hope will allow comparison to previous research and increase the generalizability of our findings. However, we did not evaluate knowledge of the guidelines and recommendations, which is a limitation as participants may not have been aware of them and this could have influenced their behaviors.

Our research design is another limitation. Our study was cross-sectional, so cause and effect cannot be interpreted from our findings. In addition, our analysis was limited to diet and physical activity only, and did not consider other characteristics or covariates that could potentially influence BMI, such as sleep quality [80], stress [81,82], and screen time [83]. Sleep quality has been found to have a unique, bidirectional relationship with BMI in which poor sleep influences weight gain, while higher BMI results in poor sleep quality [80]. The influence of stress is contradictory with some evidence indicating that stress can more than double the risk for obesity [82], while other evidence demonstrates that higher perceived stress is associated with lower BMI [81]. By comparison, the influence of screen time is straightforward and consistent. Both leisure and work-related screen time significantly increase the risk for obesity [83].

The potential effects of both self-selection bias and social-desirability bias should also be considered as a limitation. Specifically, APNs who regularly engage in healthy diet and physical activity behaviors may have self-selected to participate in the survey. Additional selection bias may have occurred due to the healthy worker effect. Individuals who remain in an occupation over time are likely healthier and participate more often in healthy behaviors than those who leave the profession. There may also be greater emphasis within advanced nursing practice to remain healthy, resulting in greater participation in behaviors known to provide health benefits. Related to this, due to the widespread recognition of the advantages of both a healthy body weight and healthy diet and exercise behavior, responses may have been influenced by social-desirability bias, with participants reporting what they believed to be the most desirable responses rather than the most accurate ones. To compensate for the potential influences of bias on our findings, we disseminated information regarding the study widely through professional colleagues and professional organizations in order to obtain the most diverse sample possible. However, we cannot exclude the possibility that bias influenced our results.

### 4.2. Future Research

Studies using objective measurement of height and weight are needed, and use of diet and exercise logs would likely improve accuracy of reporting. In addition, calculation of 24-h intake and direct assessment of metabolic rate through indirect calorimetry or handheld calorimetry would allow future researchers to control for energy balance. A stronger research design would also provide more compelling evidence. Longitudinal research could be used to evaluate trends over time and ensure greater confidence regarding the impact of exercise and diet on BMI in this population. Finally, we recommend research focusing on development and evaluation of curriculum content for physicians and APNs regarding lifestyle medicine [84] that includes emphasis on both exercise and diet behaviors.

## 5. Conclusions

The prevalence of overweight and obesity among APNs in the U.S. is less than what is reported for the general adult population. Exercise behaviors, especially participation in moderate–vigorous physical activity, have the greatest impact on BMI, and are associated with clinically meaningful differences. By comparison, healthy diet behaviors, including consumption of fruits, vegetables, and protein, are relatively widespread among APNs but do not appear to have a significant impact on BMI. APNs are patient educators and role models. Given the significant health impact of obesity, the potential influence of APNs is unquestionable.

## Figures and Tables

**Table 1 nutrients-17-03654-t001:** Participant characteristics (N = 1268).

Characteristic	Mean ± SD
Age (years)	46.7 ± 11.2
Height (cm)	165.6 ± 7.1
Weight (kg)	73.0 ± 16.0
Body Mass Index (kg/m^2^)	26.6 ± 5.4
Normal Weight	568 (45)
Overweight/Obese	700 (55)
	**Frequency (%)**
Gender (female)	1231 (97)
Race/Ethnicity *	
Asian/Pacific Islander	38 (3)
Black	32 (3)
Hispanic	61 (5)
Native American	15 (1)
White	1140 (90)
Other	14 (1)
Engage in moderate–vigorous physical activity	
Yes	580 (46)
No	688 (54)
Engage in muscle strengthening	
Yes	678 (53)
No	590 (47)
Consume fruits and vegetables	
Yes	782 (62)
No	486 (38)
Consume protein	
Yes	1197 (94)
No	71 (6)

Data reported as mean ± SD or frequency (%) * Number of responses exceeds 1268 because participants could choose more than one race/ethnicity.

**Table 2 nutrients-17-03654-t002:** Comparison of normal weight versus overweight/obese participants.

Characteristic	Normal Weight (n = 568)	Overweight/Obese (n = 700)	Z Statistic	*p*-Value
Age (years)	45.7 ± 11.0	47.5 ± 11.2	−2.88	0.004
BMI (kg/m^2^)	22.3 ± 1.7	30.1 ± 4.8	−30.66	˂0.001
Engage in moderate–vigorous physical activity				
Yes	318 (56)	262 (37)	−6.59	˂0.001
No	250 (44)	438 (63)		
Engage in muscle strengthening				
Yes	355 (63)	323 (46)	−5.81	˂0.001
No	213 (37)	377 (54)		
Consume fruits and vegetables				
Yes	368 (65)	414 (59)	−2.06	0.040
No	200 (35)	286 (41)		
Consume protein				
Yes	538 (95)	659 (94)	−0.44	0.658
No	30 (5)	41 (6)		

Comparative data from Mann–Whitney U test reported as mean ± SD or frequency (%) with overall z-value of the test and the corresponding *p*-value.

**Table 3 nutrients-17-03654-t003:** Contribution of exercise and diet behaviors to BMI.

Regression Models	B	95% CI	SE	Beta	*p*	*R* ^2^
Model A—exercise †						
Moderate–vigorous physical activity	−2.636	−3.22, −2.05	0.299	−0.242	˂0.001	0.063
Model B—exercise ‡						
Moderate–vigorous physical activity	−2.103	−2.74, −1.47	0.323	−0.193	˂0.001	
Muscle strengthening	−1.353	−1.99, −0.72	0.322	−0.125	˂0.001	0.076
Model C—exercise and diet §						
Moderate–vigorous physical activity	−2.058	−2.70, −1.42	0.325	−0.189	˂0.001	
Muscle strengthening	−1.353	−1.99, −0.072	0.322	−0.124	˂0.001	
Fruits and vegetables	−0.421	−1.02, 0.18	0.306	−0.038	0.169	0.078

† *F*(2,1242) = 42.01, *p* ˂ 0.001; ‡ *F*(3,1241) = 34.26, *p* ˂ 0.001; § *F*(4,1240) = 26.19, *p* ˂ 0.001. All models adjusted for age.

## Data Availability

The data presented in this study are available on request from the corresponding author. The data are not publicly available due to privacy and ethical reasons.

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
