# Peer review of "Impact of Diet and Exercise Behaviors on Body Mass Index of Advanced Practice Nurses in the United States"

_nutrients, 2025, doi:10.3390/nu17233654_

Round 1
Reviewer 1 Report
Comments and Suggestions for Authors
The manuscript entitled “Impact of Diet and Exercise Behaviors on Body Mass Index of Advanced Practice Nurses in the United States” addresses a relevant and contemporary topic related to lifestyle behaviors, professional role modeling, and obesity risk among Advanced Practice Nurses (APNs). The study is well organized, clearly written, and presented in accordance with the journal’s standards. However, while the subject matter is of potential interest to the readership of Nutrients, the manuscript presents significant methodological and interpretative limitations that must be addressed before further consideration for publication.
General Assessment
The manuscript's main contribution lies in describing associations between self-reported diet and exercise behaviors and BMI in a large sample of APNs. However, the originality is moderate, as similar studies have already explored these variables in healthcare professionals, including nurses. The discussion would benefit from a deeper theoretical framing on health behavior, self-efficacy, and professional modeling to strengthen its scientific relevance.
Major Comments
Study design and sampling: The cross-sectional and convenience-based recruitment design severely limits causal inference and external validity. Potential selection bias should be explicitly acknowledged and discussed, as individuals more engaged in healthy lifestyles are more likely to participate.
Measurement limitations: The use of dichotomous (“yes/no”) items to assess diet and exercise behaviors substantially limits sensitivity and statistical power. More nuanced instruments (e.g., IPAQ for physical activity or validated dietary quality indices) would provide greater analytical depth. Self-reported height and weight may introduce bias, even among highly educated participants. This issue should be further discussed beyond the brief justification provided.
Statistical analysis: The combination of non-parametric testing with multiple regression requires clarification, as assumptions of linearity, normality, and homoscedasticity may not have been satisfied. Given the dichotomous nature of predictors, logistic regression may have been more appropriate.
Interpretation of results: The manuscript overstates the clinical significance of small differences (e.g., a 2.06 kg/m² predicted reduction in BMI), which cannot be causally inferred from cross-sectional associations. The conclusion that exercise “results in clinically meaningful differences” should be reformulated to reflect correlation rather than causation. The null association between diet and BMI is superficially discussed. The authors should consider potential confounders, self-report bias, and ceiling effects in dietary behavior.
Discussion and theoretical framing: The discussion is predominantly descriptive and lacks critical engagement with behavioral theories (e.g., the Theory of Planned Behavior, Transtheoretical Model, or constructs of self-efficacy). The recommendation for curricular changes in APN education is reasonable but not directly supported by the study’s empirical evidence. This section should be reframed as a hypothesis for future work rather than a conclusion drawn from current data.
References and contextualization: The bibliography is current and appropriate, but heavily focused on epidemiological data. Inclusion of theoretical and behavioral frameworks would substantially improve the scholarly depth of the manuscript.
Minor Comments
The introduction could be shortened to emphasize the knowledge gap and research rationale more directly.
Clarify whether survey questions were pilot-tested or validated prior to use.
Explicitly discuss the possible impact of un measured confounders (e.g., work stress, shift patterns, sleep, or socioeconomic status).
Author Response
Reviewer #1
Comments and Suggestions for Authors
The manuscript entitled “Impact of Diet and Exercise Behaviors on Body Mass Index of Advanced Practice Nurses in the United States” addresses a relevant and contemporary topic related to lifestyle behaviors, professional role modeling, and obesity risk among Advanced Practice Nurses (APNs). The study is well organized, clearly written, and presented in accordance with the journal’s standards. However, while the subject matter is of potential interest to the readership of Nutrients, the manuscript presents significant methodological and interpretative limitations that must be addressed before further consideration for publication.
Thank you for your feedback and the detailed comments you provided. We have sincerely tried to address each one and hope our responses and revisions meet with your approval.
General Assessment
The manuscript's main contribution lies in describing associations between self-reported diet and exercise behaviors and BMI in a large sample of APNs. However, the originality is moderate, as similar studies have already explored these variables in healthcare professionals, including nurses. The discussion would benefit from a deeper theoretical framing on health behavior, self-efficacy, and professional modeling to strengthen its scientific relevance.
Your point is well taken, but we would argue that the population in the current study (Advanced Practice Nurses) represents a specialized group of nurses that has not been previously studied to our knowledge. Advanced Practice Nurses are clinicians with advanced degrees that are distinct from generalist registered nurses. The evidence regarding healthcare professionals primarily focuses on physicians and generalist nurses. Advanced Practice Nurses make up a small but well-educated minority of the nursing profession. As such, we believe research regarding their behaviors has merit.
Additionally, we appreciate your recommendation regarding discussion around health behavior framing, self-efficacy, and professional modeling. We have added a brief description of the Health Belief Model to the Introduction to provide a theoretical framework for the study and added content to the Discussion to expand our discussion of modeling and self-efficacy that we hope you will find to be of interest.
Major Comments
Study design and sampling: The cross-sectional and convenience-based recruitment design severely limits causal inference and external validity. Potential selection bias should be explicitly acknowledged and discussed, as individuals more engaged in healthy lifestyles are more likely to participate.
Thank you for this reminder. We had previously noted the cross-sectional design and inability to infer cause and effect as part of our Limitations, but had overlooked the potential influence of bias. We have now clearly acknowledged it as one of our limitations.
Measurement limitations: The use of dichotomous (“yes/no”) items to assess diet and exercise behaviors substantially limits sensitivity and statistical power. More nuanced instruments (e.g., IPAQ for physical activity or validated dietary quality indices) would provide greater analytical depth. Self-reported height and weight may introduce bias, even among highly educated participants. This issue should be further discussed beyond the brief justification provided.
Thank you for your interest in the format of our survey. We share your concern regarding self-report and struggled with this as a design issue. We have expanded our discussion of the limitations of self-report and specifically addressed physical activity and diet in addition to height and weight.
Statistical analysis: The combination of non-parametric testing with multiple regression requires clarification, as assumptions of linearity, normality, and homoscedasticity may not have been satisfied. Given the dichotomous nature of predictors, logistic regression may have been more appropriate.
Thank you for this question. Our decision to use multiple regression was based on our dependent variable (BMI). Use of logistic regression (binomial or ordinal) requires an ordinal or dichotomous dependent variable. Because our dependent variable (BMI) was continuous, we used multiple regression based on the primary assumption that there was one dependent variable measured at the continuous level (i.e., the interval or ratio level). Additionally, we appreciate your point about the assumptions. All assumptions were met using the procedures described for SPSS multiple regression analysis. We have added wording to our description of Statistical Analysis to report assumption testing.
- Laerd Statistics (2015). Multiple regression using SPSS Statistics. Statistical tutorials and software guides. Retrieved from https://statistics.laerd.com/
Interpretation of results: The manuscript overstates the clinical significance of small differences (e.g., a 2.06 kg/m² predicted reduction in BMI), which cannot be causally inferred from cross-sectional associations. The conclusion that exercise “results in clinically meaningful differences” should be reformulated to reflect correlation rather than causation. The null association between diet and BMI is superficially discussed. The authors should consider potential confounders, self-report bias, and ceiling effects in dietary behavior.
We appreciate your point and have tried to be as conservative as possible in our conclusion while at the same time recognizing the potential significance of our findings. In this case we are somewhat at a loss. Our conclusion was that exercise “can result in clinically meaningful differences.” We believe that based on the results of our regression model, the statement is justified. Regression provides strong predictive data regarding change and use of “can” indicates that it is a possible result, not that it causative. We would ask that you please re-read our conclusion and take our argument into consideration.
Also, we are grateful for the reminder regarding other confounders. We have now specifically noted sleep quality, stress, and screen time as potential confounders in our discussion of Limitations. As we know you are aware, there are multiple other potential confounders that from a practical perspective are too numerous to list.
Discussion and theoretical framing: The discussion is predominantly descriptive and lacks critical engagement with behavioral theories (e.g., the Theory of Planned Behavior, Transtheoretical Model, or constructs of self-efficacy). The recommendation for curricular changes in APN education is reasonable but not directly supported by the study’s empirical evidence. This section should be reframed as a hypothesis for future work rather than a conclusion drawn from current data.
Thank you. We have expanded our Discussion to include the Health Belief Model and its application to self-efficacy. We truly hope that you will find it to be of interest. Also, we have now linked our recommendation regarding curriculum content to the Health Belief Model and the influence of knowledge and self-efficacy on behavior and believe the changes provide greater support and rationale for the recommendation. We hope you will agree.
References and contextualization: The bibliography is current and appropriate, but heavily focused on epidemiological data. Inclusion of theoretical and behavioral frameworks would substantially improve the scholarly depth of the manuscript.
We appreciate your interest in this area and agree that the addition of a theoretical framework has strengthened our manuscript. Thank you.
Minor Comments
The introduction could be shortened to emphasize the knowledge gap and research rationale more directly.
We have sincerely tried to keep our Introduction as succinct as possible. It describes only obesity as a problem, our variables of interest (BMI, physical activity, and diet), and our population of interest (APNs). At your request, we have now added a brief description of the Health Belief Model to provide the theoretical framework. In response to your comment here, we reviewed and shortened our Introduction somewhat, but do not believe we can shorten it further and retain the evidence that we believe provides the rational for our study. We request your understanding in this regard.
Clarify whether survey questions were pilot-tested or validated prior to use.
We apologize that we overlooked this in our description of Methods. The questions were not validated or pilot-tested. We have now acknowledged this in our description of Measurement.
Explicitly discuss the possible impact of unmeasured confounders (e.g., work stress, shift patterns, sleep, or socioeconomic status).
Thank you for this suggestion. We have expanded our discussion in this regard to include stress, sleep quality, and screen time that we believe are the most applicable to our population. We hope that you will find it to be of interest. APNs do not routinely work off shifts due to their more specialized roles, so shift patterns are not likely to exert an influence. In addition, due to their advanced degrees and specialized roles, their socioeconomic status is relatively high, and so this would also not be a likely influence.
- Martin, B., Zhong, E. H., Reid, M., O'Hara, C., & Buck, M. (2024). A descriptive summary of the advanced practice registered nurse workforce in the united states: Targeted findings from the 2022 national nursing workforce survey. Journal of Nursing Regulation, 15(1), 4-12. https://doi.org/https://doi.org/10.1016/S2155-8256(24)00023-1
Reviewer 2 Report
Comments and Suggestions for Authors
This study offers insights into the health behaviors of Advanced Practice Nurses (APNs) and their association with BMI.
The study relies entirely on self-reported height, weight, diet, and exercise. This introduces significant risk of: Inaccurate memory regarding frequency and duration of diet and exercise. APNs, being healthcare professionals and role models, are highly likely to over-report positive behaviors (exercise, fruit/vegetable intake) and under-report their actual weight or negative dietary habits. This could artificially inflate the reported prevalence of healthy diet behaviors.
The diet and exercise assessment methods are overly simplistic, relying on just four questions.
Diet quality is complex. Simply asking about the consumption of "fruits and vegetables" and "protein foods/supplements" (with a 94% reported compliance!) ignores critical factors like portion size, caloric density, intake of refined carbohydrates, saturated fat, and sugar-sweetened beverages, which are major drivers of BMI.
Defining exercise merely as "moderate-vigorous physical activity" and "muscle strengthening exercises" without quantifying the intensity, duration, and frequency precisely provides a weak measure for association.
The study is cross-sectional (a snapshot in time), establishing only association, not causation. While the conclusion states exercise can "result in clinically meaningful differences," the data cannot prove this. Individuals with a naturally lower BMI might be more likely to engage in vigorous exercise, or a lower BMI might enable them to exercise more.
The finding that healthy diet behaviors (fruits, vegetables, protein) did not have a significant impact on BMI is counterintuitive and requires strong critical evaluation. This counterintuitive result is likely due to the crude measurement of diet (i.e., the variables measured were too simplistic to capture the real drivers of BMI) and the potential self-reporting bias. The discussion should emphasize this limitation rather than framing the finding as proof that exercise is simply superior to diet for BMI management in this population.
While BMI is a useful population-level biomarker, it is a limited measure of obesity and disease risk for individuals. The study could be strengthened by discussing or acknowledging the limitations of BMI, particularly its inability to distinguish between fat mass and muscle mass (which could be relevant given the high reported rate of muscle-strengthening exercise).
The conclusion notes a lower prevalence of overweight/obesity among APNs compared to the general U.S. population. While this supports their role as role models, the discussion should acknowledge the "healthy worker effect," where healthcare professionals may be inherently more health-conscious or subject to external pressure to maintain a healthy weight.
The presentation of the predicted decrease in BMI (2.06 kg/m2 and 1.35 kg/m2) is a key finding, but the full multivariable logistic regression model details (including p-values for the main effects, and the complete list of variables adjusted for) are missing in the summary. This information is necessary to fully assess the rigor of the "predicted decrease."
Author Response
Reviewer #2
Comments and Suggestions for Authors
This study offers insights into the health behaviors of Advanced Practice Nurses (APNs) and their association with BMI.
Thank you for your thoughtful review. We appreciate your comments and suggestions and have sincerely tried to address each one in detail. We hope you will find the revisions to be both interesting and satisfactory.
The study relies entirely on self-reported height, weight, diet, and exercise. This introduces significant risk of: Inaccurate memory regarding frequency and duration of diet and exercise. APNs, being healthcare professionals and role models, are highly likely to over-report positive behaviors (exercise, fruit/vegetable intake) and under-report their actual weight or negative dietary habits. This could artificially inflate the reported prevalence of healthy diet behaviors.
We appreciate your point and have expanded our discussion of the Limitations surrounding self-report and the potential effects of bias on our results. Specifically, we have addressed social desirability bias, which as you point out, could result in over reporting. In relation to this concern, which we share, we have also discussed the limitations of other diet and exercise tools, including both over and under reporting.
The diet and exercise assessment methods are overly simplistic, relying on just four questions. Diet quality is complex. Simply asking about the consumption of "fruits and vegetables" and "protein foods/supplements" (with a 94% reported compliance!) ignores critical factors like portion size, caloric density, intake of refined carbohydrates, saturated fat, and sugar-sweetened beverages, which are major drivers of BMI.
Your point is very well taken. We recognize the issues surrounding self-report and have tried to provide additional support for the format of our questions in our discussion of Limitations. There are acknowledged weaknesses in even the most well-validated tools. Also, we have added comparative statistics for consumption among adults in general in the United States that are consistent with the protein intake reported by APNs. Consumption rates for fruit and vegetables are lower among adults in general compared to APNs, but we would argue that increased knowledge and understanding of the health benefits of fruit and vegetable consumption may influence daily intake by APNs and that our data regarding dietary intake for both fruits and vegetables and protein are accurate.
Defining exercise merely as "moderate-vigorous physical activity" and "muscle strengthening exercises" without quantifying the intensity, duration, and frequency precisely provides a weak measure for association.
Once again we appreciate your point. Anecdotally, we have observed adults in a wide range of settings who struggle with quantification of intensity, duration, and frequency. Our observations are supported by research in this area. To address this question, we have expanded our discussion of Limitations and hope you will find it to be interesting as well as providing a satisfactory response to your concern.
The study is cross-sectional (a snapshot in time), establishing only association, not causation.
Thank you. We had previously acknowledged that in our discussion of Limitations. The statement was initially at the end of a somewhat lengthy paragraph and may have been overlooked. To emphasize its importance, we have now placed it at the beginning of the second paragraph of the Limitations section.
While the conclusion states exercise can "result in clinically meaningful differences," the data cannot prove this. Individuals with a naturally lower BMI might be more likely to engage in vigorous exercise, or a lower BMI might enable them to exercise more.
Thank you for raising this very important point. We did not mean to infer direct causation from our results. However, regression allows for prediction and we based our statement on the strength of the prediction models. We specifically used “can” in our statement to clarify that it is a possible result, not that it causative. In this case, we believe our statement is justified based on our prediction models, and hope that on reflection you will agree.
The finding that healthy diet behaviors (fruits, vegetables, protein) did not have a significant impact on BMI is counterintuitive and requires strong critical evaluation. This counterintuitive result is likely due to the crude measurement of diet (i.e., the variables measured were too simplistic to capture the real drivers of BMI) and the potential self-reporting bias. The discussion should emphasize this limitation rather than framing the finding as proof that exercise is simply superior to diet for BMI management in this population.
Your points are well taken, and we were surprised to find the absence of effect of diet. However, we would argue (as we did in our Discussion) that this was more likely due to the relatively widespread consumption of fruits and vegetables and protein in our sample. As noted in Table 2, there were no statistical differences in intake between the normal weight and overweight/obese subgroups. We have now added evidence to our Discussion to support the influence of physical activity and diet on 24-hour energy expenditure as well as evidence regarding consumption of fruits, vegetables, and protein foods among adults in the United States. Protein intake is indeed ubiquitous among adults similar to our sample, but self-reported intake of fruits and vegetables in our sample exceeded that reported for adults in general. We would attribute this to greater knowledge and understanding of the health benefits of fruits and vegetables among APNs. In response to your concern, with which we agree, we have expanded our discussion of the limitations of self-report to include collection of dietary and physical activity data.
Although we recognize that it limited our study, we chose to focus on two well-known dietary recommendations that are used for health counseling, with which we believed APNs were familiar. We absolutely agree that a more detailed dietary analysis would be desirable and have recommended that diet logs be used in future research.
While BMI is a useful population-level biomarker, it is a limited measure of obesity and disease risk for individuals. The study could be strengthened by discussing or acknowledging the limitations of BMI, particularly its inability to distinguish between fat mass and muscle mass (which could be relevant given the high reported rate of muscle-strengthening exercise).
Thank you for this suggestion. We especially appreciate your point regarding muscle mass. We have added a discussion of the limitations of BMI that we hope you will find to be of interest and applicable to our study.
The conclusion notes a lower prevalence of overweight/obesity among APNs compared to the general U.S. population. While this supports their role as role models, the discussion should acknowledge the "healthy worker effect," where healthcare professionals may be inherently more health-conscious or subject to external pressure to maintain a healthy weight.
Thank you for your thoughtful suggestion. We were unaware of this phenomenon and have reviewed it with interest. We have now added a brief discussion of the potential influences of the healthy worker effect to our discussion of bias as a limitation to our study.
The presentation of the predicted decrease in BMI (2.06 kg/m2 and 1.35 kg/m2) is a key finding, but the full multivariable logistic regression model details (including p-values for the main effects, and the complete list of variables adjusted for) are missing in the summary. This information is necessary to fully assess the rigor of the "predicted decrease."
We apologize if we have misinterpreted your comment. We believe you are referring to Table 3 that reports our regression models. We could not use logistic regression because our dependent variable (BMI) was continuous. Use of logistic regression (binomial or ordinal) requires an ordinal or dichotomous dependent variable. Because our dependent variable (BMI) was continuous, we used multiple regression based on the primary assumption that there was one dependent variable measured at the continuous level (i.e., the interval or ratio level). The model was adjusted for age, based on the statistical difference between groups noted in Table 2. The adjustment is reported in text (Section 3.4) and as a note beneath Table 3. Due to the overwhelming majority of participants reporting female gender and White race, we did not consider those variables. We hope you will agree with our decision in this case.
- Laerd Statistics (2015). Multiple regression using SPSS Statistics. Statistical tutorials and software guides. Retrieved from https://statistics.laerd.com/
Round 2
Reviewer 1 Report
Comments and Suggestions for Authors
Dear Authors,
Thank you for submitting the revised version of your manuscript and for the detailed responses to the previous review. I have carefully examined the revisions made and compared them with the issues raised in the initial round of review. Several aspects of the manuscript have been improved and now align well with the journal’s standards. However, two points still require additional attention before the manuscript can be considered for further editorial evaluation.
- Use of Causal Language
Despite the revisions, the manuscript still includes the statement that exercise “can result in clinically meaningful differences,” which implies causality. Given the cross-sectional nature of the study, causal inference is not appropriate. Please revise all such statements to reflect strictly associative language (e.g., “is associated with clinically meaningful differences”).
- Curriculum Recommendations
The recommendation regarding curriculum development remains presented as a conclusion derived from the study’s findings. This recommendation is not empirically supported by the current data and should be reframed as a potential area for future investigation rather than an implication based on the results.
The manuscript has undergone substantial improvement, and most prior concerns have been adequately addressed. However, the two remaining issues outlined above must be revised to ensure interpretative accuracy and alignment with methodological standards. I encourage the authors to revise the manuscript accordingly.
Sincerely,
Author Response
Comments and Suggestions for Authors
Dear Authors,
Thank you for submitting the revised version of your manuscript and for the detailed responses to the previous review. I have carefully examined the revisions made and compared them with the issues raised in the initial round of review. Several aspects of the manuscript have been improved and now align well with the journal’s standards. However, two points still require additional attention before the manuscript can be considered for further editorial evaluation.
Thank you for your positive feedback. We appreciate your comments and have seriously considered each of the two remaining points of concern.
- Use of Causal Language
Despite the revisions, the manuscript still includes the statement that exercise “can result in clinically meaningful differences,” which implies causality. Given the cross-sectional nature of the study, causal inference is not appropriate. Please revise all such statements to reflect strictly associative language (e.g., “is associated with clinically meaningful differences”).
Although we believe that our use of “can” softens the inference of causality, we recognize your point and agree that the inference could still be made. We have replaced “can result in clinically meaningful differences” with “are associated with clinically meaningful differences” in the Conclusions of the abstract and main body of our manuscript.
- Curriculum Recommendations
The recommendation regarding curriculum development remains presented as a conclusion derived from the study’s findings. This recommendation is not empirically supported by the current data and should be reframed as a potential area for future investigation rather than an implication based on the results.
We regret that our revised wording was still interpreted to be a conclusion. We had not intended to give that impression. We have now revised the statement altogether and placed it within the 4.2 Future Research section. We hope that you will agree with our revised statement.
The manuscript has undergone substantial improvement, and most prior concerns have been adequately addressed. However, the two remaining issues outlined above must be revised to ensure interpretative accuracy and alignment with methodological standards. I encourage the authors to revise the manuscript accordingly.
Thank you. We hope you will find our further revisions to be satisfactory.
Reviewer 2 Report
Comments and Suggestions for Authors
After seeing the manuscript's modifications, I agree with its publication.
Author Response
Reviewer #2
Comments and Suggestions for Authors
After seeing the manuscript's modifications, I agree with its publication.
Thank you again for your thoughtful analysis and feedback on our manuscript.